



# Iceberg influence on snow distribution and slush formation on Antarctic landfast sea ice from airborne multi-sensor observations

Steven Franke[1,2], Mara Neudert[2], Veit Helm[2], Arttu Jutila[3], Océane Hames[4], Niklas Neckel[2], Stefanie Arndt[2,5], and Christian Haas[2,6]

[1]Department of Geosciences, Tübingen University, Tübingen, Germany
[2]Alfred Wegener Institute, Helmholtz Centre for Polar and Marine Research, Bremerhaven, Germany
[3]Finnish Meteorological Institute, Helsinki, Finland
[4]École Polytechnique Fédérale de Lausanne, Lausanne, Switzerland
[5]University of Hamburg, Institute of Oceanography, 20146 Hamburg, Germany
[6]Institute of Environmental Physics, University of Bremen, Bremen, Germany

**Correspondence:** Steven Franke (steven.franke@uni-tuebingen.de)

**Abstract.** Antarctic landfast sea ice fringes much of the coast of Antarctica and plays an important role for coastal ice–ocean–atmosphere interaction and ice shelf stability, as well as for the sea ice associated ecosystem. It is often characterized by embedded icebergs, which influence wind-driven snow distribution and properties. Using high-resolution data from an airborne multi-sensor survey over landfast sea ice in Atka Bay, Dronning Maud Land, in December 2022, we investigate
5    the characteristics of extensive snow drifts around icebergs and their impact on flooding. An airborne quad-polarized, ultra-wideband microwave (UWBM) snow radar and laser scanner reveal persistent snow accumulation patterns around icebergs, with thick snow drifts on the windward side of icebergs, elongated lateral snow drifts parallel to the prevailing wind direction along their sides, and virtually snow-free regions with rough ice surfaces in their lee. The mass of the thick wind-facing and lateral snow drifts pushes the sea ice locally below sea level leading to flooding and slush formation at the base of the snow
10    drifts. These heterogeneous snow–water–sea-ice interfaces cause increased cross-polarized backscatter due to depolarization in the UWBM radar returns, providing a means for slush detection by airborne radar surveys. Presence of slush is confirmed by ground-based electromagnetic induction sounding data as well as with in situ measurements. Our study documents the significant influence of icebergs on snow thickness variability and redistribution over landfast sea ice and for slush formation. Moreover, it demonstrates that the snow in the lee of icebergs is thin, resulting in high radar backscatter in SAR imagery. These
15    insights improve our understanding of wind-driven snow distribution and its impact on flooding on iceberg-laden landfast sea ice, contributing to better assessments of snow transport, sea ice mass balance, and climate modeling around Antarctica.



# 1 Introduction

Landfast sea ice plays a crucial role in Antarctic coastal processes, influencing ice shelf stability, oceanographic circulation, biogeochemical cycles, and marine ecosystems, while also serving as a key indicator of climate variability and change (Fraser et al., 2023). As a semi-stationary component of the coastal cryosphere, landfast ice provides a stable platform for biological micro-organisms and fauna, affects heat and momentum exchange between the ocean and the atmosphere, and can modulate freshwater fluxes into the ocean, with implications for water mass formation and circulation (Heil et al., 1996; Lund-Hansen et al., 2024). It also plays a vital role in stabilizing ice shelves by dampening waves, providing back stress, and hampering warming of surface waters, thus mediating dynamic interactions between the ocean, ice, and atmosphere (Li et al., 2020; Fraser et al., 2012; Massom et al., 2018; Fraser et al., 2023).

Antarctic landfast sea ice is often closely linked to the presence of grounded icebergs that provide pinning points for stable ice formation and control the stability and extent of the landfast ice, particularly along the coast of East Antarctica and the Weddell Sea. Observational studies and climate models consistently show that the spatial distribution of fast ice in winter and spring is often governed by the presence and distribution of grounded icebergs rather than solely by atmospheric or oceanic conditions (Fraser et al., 2012; Li et al., 2020; Fraser et al., 2023). Trapped icebergs act as anchor points for fast-ice formation and can buffer the ice against dynamic and thermodynamic forcing (Massom et al., 2001). In regions with high iceberg concentrations, such as the eastern Weddell Sea, fast ice has been shown to be more persistent and resilient to climate-induced changes (Fraser et al., 2023). However, Coupled Model Intercomparison Project Phase 6 (CMIP6) model projections indicate a likely decline in the duration of the fast-ice season, attributed to the loss of protective pack ice, rising temperatures, and increased storm activity (Fraser et al., 2023). Despite this, fast ice in iceberg-rich zones may show some resilience to climate change, highlighting the critical role of grounded icebergs in modulating local ice conditions.

Satellite synthetic aperture radar (SAR) imagery often shows characteristic radar backscatter patterns around icebergs embedded in landfast ice sea, with extended regions of increased backscatter on one side of the icebergs. Closer inspection of the meteorological characteristics of regions with such iceberg associated backscatter patterns shows that these patterns particularly form in regions with persistent dominant wind direction, e.g. due to katabatic outflows and synoptic-scale systems, and that the high backscatter regions are located leeward of the icebergs. For example, in Atka Bay, the study region of this work (Fig. 1 and 2), the prevailing wind direction is from the East to West (in 45% of cases; Klöwer et al., 2013). There is intense debate among snow and radar experts if the high backscatter is caused by large or small snow thicknesses leeward of the icebergs (W. Dierking, A. Fraser, pers. comm.). Here we provide evidence, for the first time, that the high SAR backscatter on the iceberg lee side is related to the virtual absence of snow, revealing the original, rough sea-ice surface as observed by airborne laser scanning (ALS). Our SAR and ALS data demonstrate how icebergs embedded within landfast sea ice modulate the wind field and serve as physical barriers that alter snow redistribution and deposition patterns. The resulting heterogeneous snow distribution has critical implications for sea-ice thermodynamics and surface energy balance, as variations in snow thickness modulate thermal insulation and ice growth rates (Massom et al., 2001; Arndt et al., 2021). Areas of thick snow can



suppress sea-ice growth by insulating the ice from the cold atmosphere, while thin or bare ice regions promote enhanced basal ice accretion (Sturm et al., 1997).

Thick snow on relatively thin sea ice causes flooding of the snow–ice interface, a widespread process on Antarctic sea ice altering the bulk physical structure of sea ice and snow and affecting their energy and mass balance. Flooding leads to the formation of slush layers, i.e. seawater saturated snow with poorly bonded aggregates of rounded grains (Fierz et al.,

2009). Capillary wicking of brine into overlying snow layers has been identified as a key mechanism by which saline water infiltrates the snowpack, particularly during early stages of ice formation or after snowfall events on young ice (Massom et al., 1997, 1998). Over time, the slush layer can freeze to form snow ice. Regionally, the contribution of snow ice varies: in the Weddell Sea it accounts for around 10% of sea-ice thickness, while in the Bellingshausen and Amundsen Seas it can make up as much as 40% (Lange et al., 1990; Haas et al., 2001; Jeffries et al., 2001; Arndt et al., 2021, 2024). In our study region

of Atka Bay (Fig. 1), snow accumulation and potential flooding have been quantitatively assessed, confirming the presence of snow-ice formation and highlighting its spatial variability (Arndt et al., 2020). The presence of thick snow drifts around icebergs promotes the formation of slush and snow ice at the base of the snow.

Here, we present a detailed investigation of snow distribution and flooding patterns around icebergs in landfast sea ice in Atka Bay which is located near the German research station Neumayer III in Dronning Maud Land. Using airborne ultra-wideband

microwave (UWBM) radar data, topographic laser scanning (ALS), images of near-infrared surface radiation and subsequent ground-based data collected during the 2022/23 Antarctic season, we identify regions of high snow thickness and characteristic UWBM reflection patterns, which we interpret as flooded snow areas. We demonstrate that the distribution of snow and the occurrence of surface flooding in the landfast sea ice of Atka Bay are strongly controlled by the presence of icebergs, which drive wind-modulated snow deposition patterns. Atka Bay's persistent easterly winds combined with the topographic influence

of icebergs lead to characteristic zones of snow accumulation on the wind-facing and lateral sides of icebergs, while the lee side often remains snow-free. The accumulation zones frequently coincide with surface flooding, whereas the snow-free zones are visible as distinct high-backscatter regions in SAR imagery. By quantifying the spatial extent and frequency of such features across Atka Bay, we highlight the importance of iceberg-driven snow topography in governing snow distribution. This emphasizes the need to account for sub-grid scale processes and iceberg–snow interactions in future fast-ice models and with

the interpretation of remote sensing data.

## 2 Survey region

Atka Bay is located at the northern edge of the Ekström Ice Shelf in western Dronning Maud Land, East Antarctica, bordering the eastern Weddell Sea (Fig. 1a). The bay is bounded in the south by the Atka Ice Rise (Atkakuppelen), in the west by a spreading section of the Ekström Ice Shelf, and pinned in the east by two smaller ice rumples (Sichelryggane; Fig. 1c).

The bay's geometry facilitates the formation of a stable seasonal landfast sea ice cover, which typically breaks up during the Antarctic summer (Arndt et al., 2020). Icebergs that collect and occasionally become grounded in the bay can remain trapped







**Figure 1.** (a) Overview of Atka Bay in western Dronning Maud Land, East Antarctica, (b) a detailed aerial view of Atka Bay on 23 November 2022 (additional photos are shown in Fig. A1), and (c) an overview of the measurements within the region of interest on the landfast sea ice. The bathymetry in (a) is from Bedmap 3 (Pritchard et al., 2025), and the background map in (c) is a Sentinel-2 image from 5 December 2022.



(Fig. 2 and A1) for months or even years, provided the landfast ice does not break up. However, after landfast ice breaks up, some icebergs remain in Atka Bay and can survive multiple seasons.

Thanks to the continuously occupied Neumayer Stations since 1981 (Kohlberg and Jannek, 2007; Alfred Wegener Institut Helmholz-Zentrum für Polar und Meeresforschung, 2016), year-round scientific presence (Franke et al., 2022a) and regular summer research campaigns, Atka Bay is well-studied, particularly regarding interactions between the ice shelf, sea ice, ocean, and bathymetry (Neckel et al., 2012; Hoppmann et al., 2015a, b; Eisermann et al., 2020; Arndt et al., 2020; Smith et al., 2020; Zeising et al., 2024). Since 2010, the Antarctic Fast-Ice Network (AFIN) monitoring program has been conducted, measuring snow thickness, ice thickness, freeboard, and the thickness of underlying platelet ice at drill sites along a fixed transect in Atka Bay – monthly during winter by overwintering staff and in summer during scientific campaigns (Arndt et al., 2020; Franke et al., 2022a).

Arndt et al. (2020) reported average annual maximum values for snow and ice thicknesses along the AFIN transects between 2010 and 2018. These are $\sim 0.8\,\mathrm{m}$ for snow and $\sim 2\,\mathrm{m}$ for ice, with an average maximum freeboard of $\sim 0.01 - 0.08\,\mathrm{m}$. Seasonal variations between years and spatial differences along the transects are high. Atka Bay is also characterized by the outflow of supercooled ice shelf melt water from the ice shelf cavity, resulting in the presence of a sub-ice platelet layer. Its average maximum thickness is $\sim 3 - 4\,\mathrm{m}$ (Arndt et al., 2020).

## 3 Data and methods

Our primary dataset originates from a survey flight over Atka Bay during the ANTarctic Sea Ice and Platelet Ice Survey (ANTSI) airborne campaign (Haas, 2023) conducted with Alfred Wegener Institute's (AWI) research aircraft Polar 5 (Alfred-Wegener-Institut Helmholtz-Zentrum für Polar- und Meeresforschung, 2016) on 5 December 2022 (Fig. 1). Air temperatures ranged between -4 and -1°C during the survey. The aircraft was equipped with a quad-polarized, ultra-wideband microwave (UWBM) radar, an airborne laser scanner (ALS), the Modular Airborne Camera System (MACS), and four Global Navigation Satellite System (GNSS) antennas. Additionally, ground-based measurements, including electromagnetic (EM) induction sounding and in situ sea ice drilling, were conducted in the same time period but not on the same day. Moreover, TanDEM-X SAR satellite data were used. A sketch of the measurements used in this study in the iceberg-laden landfast sea ice setting is shown in Fig. 3. The following section describes the data acquisition, processing, and methodologies applied to all utilized datasets.

### 3.1 Aircraft GNSS data

Geodetic-grade aircraft position and altitude were determined with four NovAtel DL-V3 GNSS receivers at a sampling rate of 20 Hz. To determine the flight trajectory, we used the commercial GNSS software package Waypoint 8.90 for precise point positioning (PPP) post-processing including precise clocks and ephemerides. The accuracy of the post-processed trajectory is better than 3 cm for latitude and longitude, and better than 10 cm for altitude (Franke et al., 2022b).



**Figure 2.** Digital elevation model of Atka Bay's fast ice surface topography from ALS measurements, shown in a 2D map in (a) and (b), and in a 3D canvas in (c) with a fivefold vertical exaggeration (VE). Two UWBM profiles are represented as yellow lines in (a) and (b) and shown in more detail in Fig. 5c and 6a,b. The background map in (a,d) is a Sentinel-2 image from 5 December 2022. Panels (d) and (e) show a TanDEM-X SAR images from the same region as (a) and (c) from 12 December.



## 3.2 Airborne laser scanner data

Airborne laser scanner (ALS) data were acquired with a RIEGL LMS-VQ580 laser scanner system with a 1064 nm wavelength
(near-infrared) and a scan angle of 60°. The laser scanner was mounted underneath the floor of the aircraft. The aircraft surveyed
$\sim 360$ m above ground in the high-altitude flights and $\sim 60$ m in the low-altitude flights. The resulting scan swath width for
the high-altitude flights is about 350 m with a mean point-to-point distance of $\sim 0.5$ m. The raw laser data were combined with
the post-processed GNSS trajectory, corrected for altitude of the aircraft and calibration angles to obtain the final calibrated
georeferenced point cloud (PC) data. We used crossovers to calibrate the system and to derive the elevation accuracy of the
final georeferenced PC to be better than $0.1 \pm 0.1$ m. The final digital elevation model (DEM) with 1 m horizontal resolution
(from high-altitude flights only) was derived from the PC by using an inverse distance weighting algorithm and a 5 m search
radius. All ALS elevations are represented in absolute ellipsoidal height (WGS84) which is about 12.75 m in the study region.

To generate a consistent DEM from ALS swaths, we analyzed each individual profile strip of the ALS DEM and adjusted its
offset relative to intersecting ALS DEM strips, using a reference strip as a baseline. This approach corrects for tidal influences
that cause variations in absolute sea level during the six hours long survey. The RIEGL LMS-VQ580 laser scanner system also
provides measurements of surface reflectance at the 1064 nm laser wavelength (Hutter et al., 2023). These data were gridded
with 1 m resolution as well and complement the DEM.

Furthermore, we used the ALS DEM to compute topographic roughness derived from the largest inter-cell elevation dif-
ference between the central pixel and its surrounding eight pixels as defined in Wilson et al. (2007). Note that we generally
assume that the surface of the fast ice is relatively level, and therefore variations in snow surface elevation represent variations
in snow thickness.

## 3.3 Near-infrared radiation from MACS images

The Modular Airborne Camera System (MACS) is a multispectral, high-resolution geodetic grade nadir camera system de-
veloped by the German Aerospace Center (DLR). Its near-infrared (NIR) camera is mounted underneath the aircraft floor and
captures light within the $715-950$ nm wavelength range (Rettelbach et al., 2024). We used 11473 NIR images to generate an
orthomosaic of the study region. NIR data were acquired at a shutter speed of 0.1 ms to avoid image saturation from overex-
posure of bright air–snow interfaces. All images were corrected for vignetting effects by employing a parametric vignetting
model (Rohlfing, 2012) and converted from MACS raw data format into lossless PNG images. The external camera orientation
was obtained from the camera inertial measurement unit (IMU) and the GNSS solutions also used for the ALS processing.
The final orthomosaic was obtained by employing the full structure from motion pipeline of Agisoft Metashape (Agisoft LLC,
2022; Neckel et al., 2023).

## 3.4 Airborne ultra-wideband microwave (UWBM) radar data

The airborne ultra-wideband microwave (UWBM) radar used in this study is a quad-polarized, frequency-modulated continuous-
wave (FMCW) radar with a $2-18$ GHz bandwidth, developed by the Center for Remote Sensing and Integrated Systems (CRe-



**Figure 3.** (a) Cross-section of the landfast sea ice and sketch of the measurements used in this study, as well as aspects of iceberg-influenced sea ice in Atka Bay and the prevailing snow distribution. The drawings in the bottom row (b) – (d) illustrate the observations and interpretations of reflection characteristics in the UWBM radar data at different material transitions (e.g., bare ice, snow-to-ice transition, or dry-to-flooded snow transition).

SIS) at the University of Kansas (Yan et al., 2017). It consists of dual-polarized transmitter and receiver antennas that capture the radar echo from snow and ice surfaces, allowing measurements at all four polarizations at high precision (cm-scale). The UWBM radar was mounted under the floor of AWI's Polar 5 aircraft and operated at two different altitudes of ∼ 60 m (low-



altitude mode) and $\sim 360$ m (high-altitude mode) above ground (Fig. 1c). The low-altitude setup results in cross-track and along-track pulse-limited footprint sizes of approximately 2.6 m and 1.0 m, respectively, minimizing clutter and improving

accuracy in detecting the air–snow interface and the underside of the dry snow layer (Jutila et al., 2022a, b). The high-altitude mode increases the footprint diameter but enables larger swaths for the airborne laser scanner and larger footprints for the MACS camera.

UWBM raw radar data processing was performed with the Open Polar Radar Toolbox (Open Polar Radar, 2023) following Jutila et al. (2022a). The range sample interval of the processed radar data is 0.0706 ns, the along-track trace spacing 1.35 m,

and the spacing between UWBM profile lines is $\sim 300$ m (Fig. 1c).

### 3.5 Synthetic aperture radar (SAR)

Iceberg-influenced, wind-driven snow accumulation patterns as well as icebergs themselves represent prominent features in synthetic aperture radar (SAR) data. We therefore processed one SAR image of the survey region acquired by TanDEM-X on 12 December 2022, i.e. 7 days after the airborne survey over Atka Bay (Fig. 4b). The data were processed from Single

Look Complex ScanSAR format to terrain corrected backscatter coefficient gamma nought ($\gamma^0$, Small, 2011). The processing includes calibrating, mosaicing, and multi-looking followed by a terrain correction and geocoding step. For the latter two steps, we employed the gridded ALS DEM resulting in a final ground resolution of 15 m.

### 3.6 Ground-based electromagnetic induction sounding

Ground-based electromagnetic (EM) induction sounding measurements were carried out with a multifrequency EM sounder,

the GEM-2 instrument from Geophex Ltd with the primary objective of studying the sub-ice platelet layer (Neudert et al., 2024). Between 27 October 2022 and 3 January 2023, the GEM-2 data were collected along drill hole transects with a resolution of 10 m along-track (Neudert and Arndt, 2025) with signal frequencies of 1530, 5310, 18330, 63030, and 93090 Hz. Higher frequencies are more sensitive to conductivity changes near the surface, which makes them more responsive to the presence of saline, conductive slush resulting from flooding.

We developed a qualitative indicator to assess the presence of surface flooding using GEM-2 measurements. This indicator, referred to as the flooding score, was calculated by taking the ratio of the quadrature response at 93090 Hz to the inphase response at 18330 Hz, and normalizing it by the retrieved, total ice-plus-snow thickness. More details are described in Neudert et al. (2024). While a full multifrequency inversion that includes an explicit slush layer would provide a more detailed subsurface profile, we use this simplified flooding score as a practical alternative due to its computational efficiency and ease

of implementation. A complete multifrequency inversion approach, incorporating slush as a separate layer, is currently under review (Neudert et al., 2025, in review).

To evaluate how this score behaves under different conditions, we performed forward modeling across a range of realistic scenarios. We varied the thicknesses of ice (1.0–2.5 m), snow (0–3 m), slush (0–2 m), and the sub-ice platelet layer (0–6 m), and used different conductivity values for each layer. The modeling showed that in all cases without flooding, the flooding score

stayed within a range of 0 to 10. Higher scores were associated with the presence of flooding. Actual GEM-2 measurements





showed similar behavior, with observed flooding scores typically peaking around 6, a standard deviation of 7, and some scores reaching up to 36.

Based on these results, we defined three classification ranges: scores below 8 were labeled as no flooding, scores between 8 and 10 as possible flooding with low confidence, and scores above 10 as flooding with high confidence. To further simplify the
analysis, we assigned flooding probabilities of 0, 0.5, and 1 to these categories, respectively.

### 3.7 Sea-ice drilling

Manual drill-hole measurements of snow, sea-ice, and sub-ice platelet layer thickness, and ice freeboard were conducted across Atka Bay in the Antarctic summer season 2022/23. The measurements were carried out along an east-west transect as well as on two perpendicular north-south transects (Fig. 1c). For a detailed summary of the drilling activities in the 2022/23 season,
we refer to Neudert and Arndt (2024) and the AWI ANT-Land 2022/23 field report (Regnery et al., 2024, Section 4.6: AFIN – Antarctic Fast Ice Network).

### 3.8 Freeboard estimation from ALS data

By measuring the air–snow interface elevation, the ALS contributes critical data for estimating snow freeboard $F$, which is the height of the snow-covered sea-ice surface above the local sea level. We retrieved the average sea surface height of the fast ice
from overflights of two small open-water leads within the adjacent pack ice north of the fast ice edge providing reference points from which to derive the local sea level height (Fig. 1c). Unfortunately, the pack ice had recently closed and there was little open water. Snow freeboard was then obtained by substracting retrieved sea level height from the ALS DEM surface elevation.

### 3.9 Snow thickness determination from UWBM data and validation

We derived snow thickness $H_{snow}$ from the two-way travel time (TWT) delay between the reflection from the air–snow
interface $TWT_{surf}$ and the reflection from the snow base $TWT_{base}$. The air–snow interface reflection was traced in the co-polarized HH data (Willatt et al., 2023). It was picked where the leading edge of the local backscatter maximum has the strongest gradient. This method applies the Threshold Centre of Gravity (TCOG) algorithm (Davis, 1997) within a specific window, capturing the leading edge with a threshold of 0.8. Locations where the autotracker failed were traced manually. The snow-base reflection was traced manually at the maximum amplitude of the deepest strongly reflecting interface in the
radargrams.

The one-way travel time delay $\Delta t = \frac{1}{2}(TWT_{base} - TWT_{surf})$ between these two reflections was converted into snow thickness $H_{snow}$ by multiplying it with the radar wave propagation speed in snow $v_{snow}$:

$$H_{snow} = \Delta t \cdot v_{snow}. \tag{1}$$



We derived the radar wave speed in snow by applying $v_{snow} = \frac{c_0}{\sqrt{\varepsilon'_{snow}}}$, where $c_0$ is the speed of light in vacuum and $\varepsilon'_{snow}$
is the relative dielectric permittivity of snow. By following Kovacs et al. (1995), we derived the $\varepsilon'_{snow}$ which depends on the
snow density $\rho$:

$$\varepsilon'_{snow} = (1 + 0.851\rho)^2. \tag{2}$$

Hoppmann et al. (2012) measured snow densities in August 2011 over multiple locations in Atka Bay ranging between $0.33 -$
$0.45\,\mathrm{g\,cm^{-3}}$ with a mean density of $0.38\,\mathrm{g\,cm^{-3}}$. Based on these values, we used an average snow density of $0.38 \pm 0.06\,\mathrm{g\,cm^{-3}}$,
leading to $\varepsilon'_{snow}$ = 1.75 and $v_{snow}$ = $2.27 \cdot 10^8\,\mathrm{m\,s^{-1}}$ to derive the snow column thickness from UWBM reflections and used
a density uncertainty of $0.06\,\mathrm{g\,cm^{-3}}$ to determine snow column thickness uncertainties $\Delta H_{snow}$ ($\Delta H_{snow} \approx 0.04\,\mathrm{cm}$ per cm
snow column thickness). We justify our approach by the fact that we obtain nearly identical values for the electromagnetic
wave propagation speed as a function of snow density using the method of Hallikainen et al. (1986).

Since the UWBM data were acquired over sea ice with areas of potential flooding due to a critical snow load, we specifically
define our snow column thickness as dry-snow thickness. The dry-snow depth represents either the interface between snow and
ice or the interface between dry snow and flooded snow (i.e., slush). Due to wind-driven accumulation differences, variations
in snow liquid water content across Atka Bay, and refreezing of slush, and differences in snow compaction, the actual radar
propagation speed may exhibit lateral variations. Note that with the relatively high air temperatures of between -4 and -1°C
during the survey, near-surface snow liquid water content may have been slightly risen.

To validate our UWBM-derived snow thickness calculations, we compared these values with manually measured snow
thickness measurements at drill holes (white circles in Fig. 1c) within the AFIN measurements (Regnery et al., 2024) which
were conducted at the same time of the UWBM survey.

### 3.10 Dry-snow base return power determination

We calculate the return power of the dry-snow base reflection (the strength of the radar signal reflected from the interface
between dry snow and ice or flooded snow) in decibel $P\,[\mathrm{dB}]$ for each polarization (HH, VH, VV, HV) at every radar trace ($t$)
at the range of the interface pick ($r$) by applying a simplified approach as in e.g., Jordan et al. (2016) and Franke et al. (2021)
by integrating the backscatter above and below the interface pick within an envelope of $\pm$ 10 range bins:

$$P\,[\mathrm{dB}] = \int_U^L P(t,r), \tag{3}$$

where $U$ represents the upper limit (10 range bins above the reflection) and $L$ the lower limit (10 range bins below the
reflection). Note that a window of 20 range bins (0.07 ns per range bin) corresponds to a total TWT of approximately 1.4 ns
and a vertical distance of 32 cm, using the electromagnetic wave propagation speed in snow applied in this study. We consider
this time window to be sufficiently long to capture scattering arriving in the TWT domain above and below the maximum
amplitude of the reflecting interface as well as volume scattering a few centimeters in range around the picked interface, yet





short enough to exclude reflections from other sources. Due to differences in the footprint between high- and low-altitude

240 mode UWBM reflections, we expect different characteristics particularly for volume scattering making interface return power

comparisons between these two modes ambiguous. Therefore, we only use high-altitude mode flights for the determination of

interface return power.

**(a)** Surface roughness from ALS DEM

Topographic roughness
Low — High

**(b)** TanDEM-X SAR

Backscatter
Low — High

**(c)** Laser scanner reflectance (active)

Reflectance
Low — High

0    2.5    5 km

**(d)** Near-infrared radiation (passive)

Radiation
Low — High

**Figure 4.** (a) Surface roughness derived from the ALS DEM, (b) backscatter from a TanDEM-X radar image, (c) ALS reflectance, (d) near-infrared radiation from MACS images.





## 4 Results

### 4.1 Snow distribution and surface properties of iceberg-influenced landfast sea ice

The laser scanner DEM shows prominent characteristics of the snow distribution around icebergs. On their wind-facing sides, a pronounced snow drift forms, reaching heights of up to 5 m above the surrounding surface (Fig. 2). The thickness of these snow drifts gradually decreases with increasing distance in the direction toward the prevailing wind direction (here toward the east). Additionally, two lateral snow drifts are observed on the left and right sides of icebergs, and slightly leeward, which are lower in height compared to the wind-facing snow drift. The lateral snow drifts extend for several hundred meters along

the wind direction. The laser scanner DEM also shows that a broad, low-elevation region extends for several hundred meters to several kilometers leeward of the icebergs and between the lateral drifts (Fig. 2a – c). When multiple icebergs are closely spaced, their individual wind-induced snow distribution patterns overlap, forming a complex system of high snow drifts and low-lying areas. There is also a prominent, narrow wind scoop several meters deep between the icebergs and their wind-facing snow drifts.

These distinct snow elevation patterns created by the icebergs coincide closely with the radar backscatter patterns in the SAR images (Fig. 2d, e), providing compelling support for their interpretation. The leeward high-backscatter regions coincide with regions of thin snow and relatively large topographic roughness arising from small-scale surface variations in the ALS DEM (Fig. 2d,e and 4a,b). The leeward regions also show pronounced reductions of the ALS reflectance and MACS near-infrared radiation data compared to the surrounding regions with more snow (Fig. 4c, d). In addition, thick wind-facing and lateral snow

drifts also coincide with increased backscatter regions in the SAR data.

**Table 1.** Comparison of snow depth $H_{snow}$ and freeboard $F$ at AFIN drill sites with UWBM-derived snow depth and HH/VH ratio. Furthermore, the distance between the closest UWBM trace and AFIN site and date of drilling is shown.

| AFIN Site | Date (drill) | $F$ (drill) | $H_{snow}$ (drill) | $H_{snow}$ (UWBM) | HH/VH | Distance |
|---|---|---|---|---|---|---|
| ATKA 03 | 2022-12-05 | 2 cm | 58 cm | 57 $\pm$ 2 cm | 1.2 | 19 m |
| ATKA 07 | 2022-12-05 | 11 cm | 40 cm | 47 $\pm$ 2 cm | 1.4 | 27 m |
| ATKA 11 | 2022-12-05 | 2 cm | 59 cm | 62 $\pm$ 2 cm | 1.2 | 37 m |
| ATKA 21 | 2022-12-05 | -21 cm | 108 cm | 104 $\pm$ 4 cm | 1.1 | 102 m |
| AFIN 02 SNw | 2022-12-13 | -17 cm | 119 cm | 122 $\pm$ 5 cm | 1.1 | 9 m |

### 4.2 UWBM radar reflection characteristics

Using the UWBM radar, we analyzed the snow surface elevation patterns around the icebergs by means of vertical cross-sections in all four polarizations (HH, VH, VV, HV). For a general characterization of the structural reflection properties, we



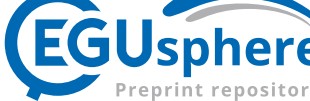

**(a)** **(b)**

**(c)**

**(d)**

**Figure 5.** Reflection characteristics in UWBM radargrams (HH) of snow-covered fast ice near an iceberg. Panels (a) and (b) present 3D canvases in which the topography around the iceberg (vertically exaggerated (VE) by a factor of five) is shown along with the locations of the UWBM profiles (a), as well as the corresponding elevation-corrected radargrams (b). Two profiles are highlighted, and their radargrams are displayed in panels (c) and (d). The elevation axis is referenced to the WGS84 ellipsoid and the local sea level is indicated by a horizontal white dashed line.





initially focus on HH-polarized reflections only. In addition, we compare co- and cross-polarized reflection patterns for HH
and VH (HH is when signal transmission is in H and acquisition in H; VH is when signal transmission is in H but acquisition
in V) to study the effect of radar wave depolarization.

In the UWBM radargrams, we identify three distinct types of reflections beneath the air-snow interface (examples are demonstrated in Fig. 6):

- *Type 1*: no additional detectable reflections below the surface return,

- *Type 2*: low backscatter reflections occurring within a few nanoseconds in range below the surface return with weak
radar wave depolarization, and

- *Type 3*: high backscatter reflections appearing within a range of approximately 10 to more than 50 ns below the surface
return with strong radar wave depolarization.

Regions where no additional reflections (*Type 1*) are detected beneath the surface return are primarily found leeward of icebergs, i.e. in areas where the ALS DEM indicates very low surface elevations. In contrast, deep reflections with high backscatter
(*Type 3*) mostly coincide with the locations of the thickest snow in the large snow drifts. Shallow reflections with low backscatter (*Type 2*) are observed in regions with moderate surface elevations, i.e. moderate snow thickness, respectively.

We assume that the continuous reflections in the UWBM data represent either the transition from snow to ice or the transition
from dry snow to flooded snow, which could be wet snow / slush or refrozen, salty snow ice. Therefore, we refer to these
reflections as the dry-snow base in both cases. In a few regions, however, and exclusively within the snow drifts near the
icebergs, we observe multiple continuous reflections (e.g. Fig. 5c, d). In these cases, we assign the upper reflections to internal
snow layers, while the lowermost, strong reflection must correspond to the dry-snow base.

## 4.3 Dry-snow depth inferred from UWBM reflections and validation at drill sites

The calculated snow thicknesses ($H_{snow}$), derived from the time delay between the air-snow reflection and the dry-snow base
reflection, are shown in Fig. 7a. Notably, snow thickness could not be derived across large portions of the study area, as
no second reflection below the surface return was detected in the radargrams at these locations. These gaps are particularly
prominent leeward of the icebergs, though not exclusively.

The zoomed-in views in Figs. $7a_{1-4}$ show that snow thickness correlates with elevated surface features in the ALS DEM
surrounding the icebergs, reaching over 5 m in snow thickness in some locations. Manual snow thickness measurements,
conducted close to UWBM profiles on the same day, show good agreement with these estimates (Table 1). However, at greater
snow thicknesses, the measured snow depth minus the freeboard appears to be lower than the detected dry-snow thickness.
This discrepancy is probably due to larger snow compaction and density of the thicker snow, which reduces the average radar
wave propagation speed, leading to an overestimation of snow thickness in radar-derived measurements (e.g., the apparent
downward-bending of the dry-snow depth reflection in the middle of Fig. 5c).







**Figure 6.** Panels (a) and (b) show two radargrams in HH and VH polarizations, whose profile locations are indicated in Fig. 2a,b, as well as the ratio between the respective HH and VH component. The surface reflection of the radargrams has been vertically aligned to 5 ns in (a) and 10 ns in (b) leading to a flattened air–snow interface. Panel (a) presents a radargram from the eastern part of our survey region, covering an area with low snow thickness as well as snow drifts adjacent to icebergs. Panel (b) shows a radargram oriented perpendicular to the predominant wind direction, traversing the wind-facing snow drift of an iceberg (the same radargram as the elevation-corrected radargram in profile c' – c" in Fig. 5a,c). The radargrams in (a) and (b) also contain examples of *Type 2* and *Type 3* reflections. Panel (c) shows correlations among $P_{HH}/P_{VH}$ ratio, snow thickness $H_{snow}$ and color-coded interface return power $P_{HH}$ and $P_{VH}$ as well as the interface elevation (referenced to the WGS84 ellipsoid). Note that from left to right the third and fourth plots are identical but in the third plot high values are plotted on top of low values and in the fourth plot low values on top of high values.



**Figure 7.** Panel (a) with sub-panels ($a_{1-4}$) show UWBM-derived snow thickness $H_{snow}$, panel (b) with sub-panels ($b_{1-4}$) show the ratio of co-/cross-polarized backscatter from the dry-snow base (here HH/VH), and panel (c) with sub-panels ($c_{1-4}$) shows GEM-2-derived flooding probability scores. The background image is the ALS-derived surface elevation and contour of the ice-shelf edge.





### 4.4  Co- and cross-polarized signals in UWBM data

A comparison of co-polarized (HH, VV) and cross-polarized (HV, VH) radargrams reveals the following characteristics. The surface reflection is significantly weaker in the cross-polarized radargrams than in the co-polarized ones. This pattern is also observed in the shallow *Type 2* reflections. However, for *Type 3* reflections, the backscatter strength is similar in both co- and cross-polarized channels, suggesting an increased depolarization along the interface of these reflections. A comparison of two representative regions where this effect is particularly pronounced is shown for HH and VH polariztions in Fig. 6a, b.

A systematic analysis of the relationship between co- and cross-polarized backscatter shows a clear correlation with snow thickness and the absolute vertical position of the reflecting interface. Figure 6c presents the ratio between co-/cross-polarized backscatter (HH/VH) at the reflecting interface, snow thickness, as well as the return power at the interface (again for HH and VH; the two leftmost panels in Fig. 6c). Additionally, it displays the calculated height of the interface in ellipsoidal height (WGS84), where 12.75 m represents the local sea level (the two rightmost panels in Fig. 6c).

This comparison highlights the following characteristics: (1) The ratio between HH/VH is determined by the cross-polarized component (VH) as all reflecting interfaces in our study show strong specular reflections but vary significantly in their volume scattering strength. (2) The cross-polarized backscatter increases with snow thickness. (3) High co-/cross-polarized backscatter ratios (blue colors in Fig. 6c) and low snow thicknesses correlate with an interface height above sea level ($> 12.75$ m), whereas low co-/cross-polarized backscatter ratios and high snow thicknesses correlate with lower interface heights (yellow colors in Fig. 6c). It is important to note, however, that for high snow thicknesses, the actual interface elevations are likely slightly higher than shown in the figure due to the compaction of the thicker snow, which increases its density and thereby reduces the EM wave propagation speed.

In Fig. 7b, we show the spatial distribution of the HH/VH ratio. It is evident that low values correlate with the snow drifts surrounding the icebergs and are also found in regions with low surface elevations (Figs. $7b_{1-4}$).

### 4.5  Flooding probability from ground-based EM induction sounding measurements

As explained above, the GEM-2 data provide information on flooding probability. The profiles cover a larger area and greater length than the UWBM profiles but are limited to regions that could be safely accessed by snowmobile. Figure 7c categorizes flooding probability into three classes: (1) no flooding, (2) flooding with low confidence, and (3) flooding with high confidence.

A comparison with the ALS DEM reveals that high flooding probabilities are detected in regions with higher elevations in the DEM, which correspond to thick snow drifts. Once again, proximity to icebergs is associated with a higher frequency of flooding detections (Figs. $7c_{1-4}$). Furthermore, the lee sides of icebergs are free of flooding detections, as are other regions with very low surface elevations, i.e. thinner snow.





# 5 Discussion

## 5.1 Effect of icebergs on snow distribution and surface roughness

Our results show that icebergs strongly modify the distribution of snow on landfast sea ice, depending on wind direction and drifting snow events. They create large snow drifts windwards of and next to themselves, and leave an almost snow-free zone in their lee, which can extend for several kilometers. These snow drift patterns significantly influence surface roughness, radar backscatter, and optical reflectance. A spatial analysis of our UWBM-derived snow thickness measurements shows that, on average, snow surface elevation and thickness are considerably larger close to icebergs compared to the average regional conditions, and decrease away from the icebergs (Fig. 8). For instance, the mean snow thickness within $100\,\text{m}$ of icebergs is more than three times higher compared to the mean snow thickness elsewhere in Atka Bay.

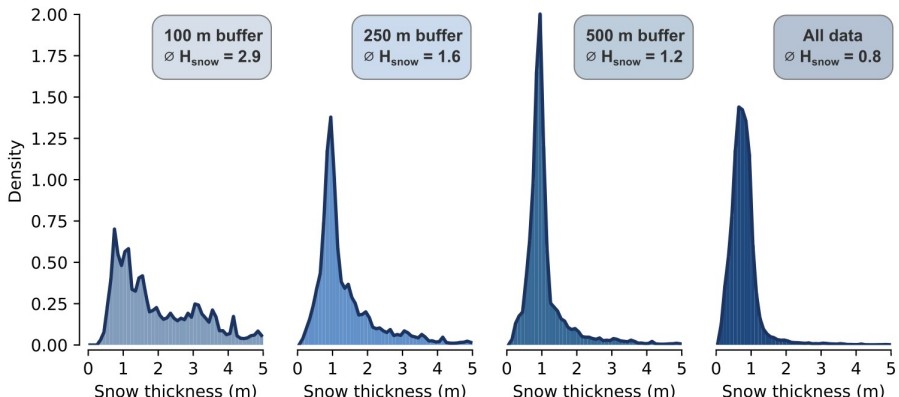

**Figure 8.** Histograms showing the spatial distribution of UWBM-derived snow thickness ($H_{snow}$) with respect to the distance from icebergs (within a $100\,\text{m}$, $250\,\text{m}$, and $500\,\text{m}$ buffer and all data). The vertical axis shows probability density, and the mean snow thickness values are provided on the top of each histogram. Note that the statistical analysis is only valid for those locations where snow thickness was detected in the UWBM and does not consider locations where snow thickness is zero.

## 5.2 Interpretation of reflection characteristics of UWBM data

Using the four polarization combinations of the UWBM (HH, VH, VV, HV), we can analyze the reflection properties at interfaces and within snow and ice (Fig. 3). We expect strong co-polarized backscatter in HH and VV where sinlge, specular reflections dominate, e.g., at the air–snow and snow–ice interfaces or at the interface between dry snow and slush or very saline ice. The air–snow and snow–ice interfaces exhibit strong reflections due to the significant contrast in dielectric permittivity (Geldsetzer and Yackel, 2009). Cross-polarized backscatter strength in VH and HV is expected to be lower at these interfaces unless the interface roughness is high (Gupta et al., 2013) or the bulk snow or ice volume has heterogeneous dielectric properties on the scale of the radar wavelengths.





Furthermore, we expect strong cross-polarized (VH and HV) backscatter where volume scattering and depolarization dominate (Scharien et al., 2012; Hajnsek et al., 2021). Volume scattering should be particularly pronounced at the transition between dry and wet snow in the presence of slush. This transition is characterized by a strong dielectric contrast due to the presence of salt water, which increases permittivity (depending on wetness) and conductivity (depending on salinity). Additionally,

the infiltration of brine into the snowpack through brine wicking – i.e., the upward migration of saline water into overlying snow via capillary forces – further alters the electromagnetic properties of the snow at the dry-snow base. This process can lead to vertical salinity gradients even within otherwise "dry" snow layers, effectively enhancing dielectric heterogeneity and contributing to volume scattering and depolarization (e.g., Massom et al., 1997). Furthermore, it exhibits a complex internal structure as well as a larger scattering path further contributing to volume scattering and multiple reflections due to varying

liquid water content and microstructure irregularities causing significant depolarization (Evans, 1965; Sihvola et al., 1985).

It is very likely that regions where no reflection other than the surface return is detected (*Type 1* reflections) correspond to areas with no snow cover (or a snow thickness too small to be resolved in the radargrams). This assumption is further supported by the fact that these regions exhibit very low surface elevations (Fig. 2) and low optical surface reflectance (Fig. 4c, d), which is indicative of an air–ice interface.

*Type 2* reflections, which appear at shallow depths below the surface return, are characterized by moderate backscatter strength in co-polarized channels and significantly lower backscatter strength in cross-polarized channels (Fig. 6a, b). This suggests a transition between two homogeneous bulk dielectric media (within the UWBM wavelength range) with low surface roughness at the interface, most likely corresponding to the snow–ice horizon. This interpretation is further supported by the fact that these interfaces with low co-/cross-polarized ratios are systematically located above sea level (Fig. 6c) and are

associated with thin snow layers and low elevations in the ALS DEM.

The large amplitudes of the deeper *Type 3* reflections, observed in both co- and cross-polarized channels, indicate an interface with a high dielectric contrast to dry snow as well as a significant amount of depolarization, suggesting strong volume scattering. Since this type of reflection is exclusively found in connection with high snow drifts around icebergs, it is most likely associated with the transition from dry to flooded snow.

Sea ice flooded by seawater forms slush, a mixture of water, snow grains with varying densities. When the slush refreezes, a layer of ice develops on top of it, growing downwards and leaving slush with increasing brine salinities below. This creates highly heterogeneous dielectric properties within the scattering volume. The increasing salinity of the basal slush increases both its electrical conductivity and relative permittivity, creating a high dielectric contrast to the dry snow above, but also leads toincreased radar wave attenuation at the same time. The strong depolarization due to high volume scattering can be well

explained by this material transition.

The interpretation that areas of small HH/VH backscatter ratios are flooded is further supported by their thick snow cover, which pushes the sea ice surface below sea level, and the fact that the thermodynamically grown ice is expected to vary little across Atka Bay (Arndt, 2022). Additionally, these locations show clear flooding detections in the GEM-2 data, and the inferred radar interface heights coincide with sea level (Fig. 6c), accounting for the effect of increasing average snow density on the

conversion between travel time and depth.



### 5.3 Implications for remote sensing and modeling of snow-covered iceberg-laden landfast sea ice and its role in the cryosphere

Our work demonstrates the promising capability of polarimetric radar for enhancing the discrimination of snow on sea ice. In Atka Bay, we detect not only the snow–ice interface (Stroeve et al., 2020; Willatt et al., 2023) but also transitions to snow-
covered flooded regions and internal interfaces suggesting multiple wind-induced snow deposition events. Incorporating such polarimetric observables into future airborne surveys could improve retrieval of snow thickness and material composition over heterogeneous Antarctic sea ice.

We show that increased backscatter in the lee of icebergs in SAR images coincides with the absence of snow that reveals the underlying, rough ice surface. This recognition is crucial for correctly interpreting SAR imagery and for mapping snow-
covered and snow-free regions of landfast sea ice at larger spatial scales. Noting that such characteristics only develop in the presence of persistent winds, while changing wind directions would blur clear snow distribution patterns with time, our findings suggest that the observed SAR backscatter patterns could be used to map prevailing wind regimes and snow transport pathways in other regions of Antarctica. If validated across different regions, this approach could improve circum-Antarctic assessments of wind variability and change as well as snow redistribution processes and their role in the surface mass balance of landfast
sea ice.

However, to fully understand the redistribution of wind-transported snow, the size and shape of icebergs must be considered. Different iceberg geometries create varying wind flow patterns, leading to distinct accumulation and erosion zones. Modeling these effects using atmospheric wind data and taking into account the topographic structure of icebergs as well as speed and frequency of wind events (Hames et al., 2025) would improve predictions of snow deposition, flooding probability and extent
with respect to iceberg size and distribution. Understanding these processes is crucial for refining climate models and improving the accuracy of remote sensing interpretations.

The interaction between icebergs, wind-driven snow transport, and flooding processes has important consequences for landfast sea-ice stability and its role in the Antarctic cryosphere. Thick snow drifts around icebergs contribute to local sea-ice submergence and flooding, which in turn affects ice mass balance, surface albedo, and the potential formation of snow ice
(Maksym and Markus, 2008; Jeffries et al., 2001; Massom et al., 2001). In addition, significant snow-free regions at the lee-side of icebergs also effect surface albedo and radiation transfer through sea ice. These processes are particularly relevant for climate models that seek to incorporate snow redistribution and flooding dynamics. Moreover, the future of iceberg-influenced landfast sea ice is uncertain in a warming climate (Fraser et al., 2023). Recent studies suggest that climate change may initially increase iceberg calving rates due to enhanced ice shelf melting (Liu et al., 2015; Greene et al., 2022), but over time, thinner
and fewer ice shelves could result in fewer grounded icebergs capable of stabilizing landfast ice. In addition to a potential decrease in fast ice area (Fraser et al., 2023), this could alter snow accumulation patterns and either increase or reduce the heterogeneity of the landfast ice snow cover.



## 6 Conclusions

Our study demonstrates the significant influence of icebergs on wind-driven snow distribution on landfast sea ice. We provide, for the first time, quantitative information about the resulting snow drift patterns, with more than 5 m thick wind-facing and lateral snow drifts upwind and to the sides of icebergs, respectively, and virtually snow-free conditions in the lee, extending leeward for hundreds of meters up to several kilometers. The variable snow distribution on otherwise level sea ice provides ideal conditions for studying the performance and capabilities of multiple airborne sensors, in particular our UWBM snow radar and NIR camera. The thick snow drifts also caused the unambiguous formation and presence of slush which could be well detected by the radar by way of strong reflection amplitudes and radar wave depolarization. With its vicinity to the Neumayer III station, Atka Bay provides ideal conditions for ground-based validation measurements. Here, in particular retrievals of slush probability from ground-based, multifrequency EM measurements were in good agreement with the spatial distribution of slush from the airborne UWBM radar.

The results have important implications for remote sensing of Antarctic landfast sea ice, as satellite SAR imagery captures the distinct backscatter signatures of iceberg-induced snow distribution patterns. In particular, we have shown that the high backscatter in the lee of icebergs is related to the absence of a significant snow cover which exposes the original rough ice surface that has formed in the initial stages of freeze-up of Atka Bay. In addition, we have gained confidence that our UWBM snow radar is capable of detecting slush on sea ice elsewhere. This opens new opportunities for mapping slush on pack ice as well where its occurence may be much more frequent than on fast ice and where its role for the sea ice mass balance is potentially larger.

We are hopeful that our results can be used for improving modeling of snow distribution around natural obstacles as they provide valuable observations of snow drifts around icebergs with various geometries, under steady prevailing winds (Hames et al., 2025). The observed snow distributions provide important information about wind and turbulence structure around icebergs. We conclude that icebergs mostly cause preferential deposition (Lehning et al., 2008) by impeding the undisturbed drift of snow, i.e. by slowing air flow upwind, and by shielding their leeward regions from incoming snow transport, while causing turbulence that prevents the accumulation of new snow from precipitation. Understanding snow distribution mechanisms on iceberg-influenced fast ice is crucial for improving climate models that incorporate snow redistribution and flooding dynamics, both of which affect ice mass balance, surface albedo, and sub-ice ecosystem processes.

*Code and data availability.* The OPR Toolbox (Open Polar Radar, 2023) is available at https://gitlab.com/openpolarradar/opr/. UWBM radar data of the ANTSI campaign over Atka Bay landfast sea ice will be made available on PANGAEA (Haas et al., 2025, https://doi.org/10.1594/PANGAEA.981151). Snow thickness and interface return power for all four UWBM polarizations will be made available on PANGAEA. The ALS-derived digital elevation model, topographic roughness and reflectance map will be made available on PANGAEA (Helm et al., 2025, https://doi.pangaea.de/10.1594/PANGAEA.982610). The near-infrared radiation mosaic derived from MACS images will be made available on PANGAEA (Neckel et al., 2025, https://doi.pangaea.de/10.1594/PANGAEA.982716). GEM-2 ice and platelet total thickness measurements from the 2022/23 AFIN summer campaign are available on PANGAEA (Neudert and Arndt, 2025, https://doi.org/10.1594/





PANGAEA.968041). GEM-2-derived flooding probabilities will be made available on PANGAEA. Data of thickness and properties of sea ice and snow of landfast sea ice in Atka Bay from ground-based drilling of the AFIN programme in November and December 2022 are available on PANGAEA (Neudert and Arndt, 2024, https://doi.org/10.1594/PANGAEA.968459). The field report for the ANTSI campaign is available at PANGAEA (Haas, 2023, https://download.pangaea.de/reference/118209/attachments/WeeklyReport_3_ANTSI2022.pdf)





**Figure A1.** Aerial photographs of icebergs in Atka Bay on 12 December (a) and 14 December 2018 (b), and 12 December 2022 (c). All photos were taken by C. Haas (AWI).

*Author contributions.* Christian Haas, Steven Franke and Mara Neudert conceptualized the study. Steven Franke wrote the manuscript with contributions from Mara Neudert, Christian Haas and Niklas Neckel. Veit Helm and Steven Franke processed UWBM data with contributions from Arttu Jutila. Mara Neudert processed GEM-2 data and acquired surface drilling data. Veit Helm processed laserscanner data and Niklas Neckel processed MACS aerial photographic data as well as TanDEM-X data. Stefanie Arndt is leading the sea ice measurement program at Neumayer Station and led the ground-based campaign in Atka Bay during the austral summer season 2022/23. All authors revised and
commented on the manuscript.



*Competing interests.* Christian Haas is a member of the editorial board of *The Cryosphere*. The authors declare no other competing interests.

*Acknowledgements.* We thank the AWI polar aircraft technicians Cristina Sans Coll and Clemens Gollin for their support in the field during the 2022/23 UWBM radar flights as well as the Kenn Borek flight crew Alan Gilbertson (Captain), Noah Hladiuk (First Officer) and Brad Friesen (AME). Logistical support in the field for the airborne radar campaign was provided by the Neumayer III (Germany) and Troll stations

(Norway). Furthermore, we are grateful for the logistics support and infrastructure for the ground-based survey in Atka Bay provided by AWI and the welcoming staff at the Neumayer III Station. We thank Jölund Asseng from the Neumayer Geophysical Observatory as well as the wintering crews of the 42$^{nd}$ and 43$^{rd}$ wintering for excellent field support and their team efforts contributing to the AFIN monitoring program. Moreover, we thank Christine Wesche for providing TanDEM-X SAR data.

For the ANTSI 2022/23 airborne campaign, we acknowledge support via the AWI funding grant AWI_PA_02135. Financial and logistical

support for the ground-based field work during the Antarctic field season 2022/23 was provided via the AWI funding grant AWI_ANT_27. Furthermore, we acknowledge the use of software from Open Polar Radar generated with support from the University of Kansas, NASA grants 80NSSC20K1242 and 80NSSC21K0753, and NSF grants OPP-2027615, OPP-2019719, OPP-1739003, IIS-1838230, RISE-2126503, RISE-2127606, and RISE-2126468. The authors would like to thank Aspen Technology, Inc. for providing software licenses and support. TanDEM-X images were provided as part of the project OCE3146.

Steven Franke was funded by the Walter Benjamin Programme of the Deutsche Forschungsgemeinschaft (DFG, German Research Foundation; project number 506043073). Mara Neudert was supported by an AWI Inspires PhD fellowship. Niklas Neckel was funded by the Bundesministerium für Bildung und Forschung (BMBF, German Federal Ministry of Education and Research) project *IceSense* (03F0866A). Arttu Jutila was supported by the Research Council of Finland project *CROS-Arctic* (341550) as well as by the BMBF project *IceSense* (03F0866A). Océane Hames was supported by the Swiss National Science Foundation through projects 200020_215406 and 200020_179130,

as well as by the AWI ANT-LAND project *SnAcc* (AWI_ANT_27). Stefanie Arndt was supported by DFG projects *fAntasie* (AR1236/3-1) and *SnowCast* (AR1236/1-1) within its priority program "Antarctic Research with comparative investigations in the Arctic ice areas" (SPP1158), and the DFG Emmy Noether Programme project *SNOWflAke* (project number 493362232).



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
