# Peer review of "Iceberg influence on snow distribution and slush formation on Antarctic landfast sea ice from airborne multi-sensor observations"

_EGUsphere, 2025_

## Referee Comment (RC1)

240  mode UWBM reflections, we expect different characteristics particularly for volume scattering making interface return power comparisons between these two modes ambiguous. Therefore, we only use high-altitude mode flights for the determination of interface return power.

---

## Author Comment (AC1)

RC1: 'Comment on egusphere-2025-2657', John Yackel, 14 Jul 2025

Dear Authors and Handling Editor,

Iceberg influence on snow distribution and slush formation on Antarctic landfast sea ice from airborne multi-sensor observations by Steven Franke and co-authors presents a highly novel multi-sensor remote sensing assessment of of iceberg infiltrated snow-covered landfast seasonal sea ice in Atka Bay, Antarctica focused on early December 2022 during the ANTSI campaign. The datasets consist of quad-polarized, ultra-wideband microwave (UWBM) radar from CReSiS U Kansas, airborne laser scanner (ALS), the Modular Airborne Camera System (MACS), four Global Navigation Satellite System (GNSS) antennas, TanDEM-X band SAR imagery, along with coincident ground-based measurements including electromagnetic (EM) induction sounding (GEM2) and in-situ snow depth and sea ice drilling conducted reasonably close in time to the remote sensing data.

The research approach and its datasets are, in my opinion, highly novel and unique. The manuscript is extremely well written and organized and includes some of the most exquisitely constructed illustrations I have seen in a long time. Figure 3 is one such example. This manuscript makes a strong contribution towards improved understanding of snow processes on Antarctic sea ice including the interpretation and use of FMCW and X-band SAR and their polarization capability for snow depth estimation. I recommend publication subject to minor revisions and addressing my questions below.

> We gratefully thank John Yackel for the time and effort reviewing our manuscript as well as for the positive and helpful feedback and comments. We outline below how we intend to address the comments and how we will modify our manuscript accordingly.

General Comment:

1) I found the Introduction written oddly in the sense that results/conclusions are alluded to on several occasions (L44-47; L67-72) without having read an objectives statement. I strongly recommend that the Introduction include explicitly written and tractable objectives statements and also remove the suggestive language as to what the results and conclusions of the analysis will show.

> We agree and will rewrite the introduction section focusing on removing suggestive statements with respect to the results or conclusions of our manuscript. Additionally, we will add the objectives of this study in the introduction.

2) The Venturi effect or similar fluid dynamic principles appear to be operating here. I suggest the authors research and possibly mention this principal and relate the effect

to blowing snow around obstacles such as icebergs and discuss whether or not they expect the wind speed to increase leeward of the icebergs further enhancing wind scouring to keep the snow cover thin.

The reviewer raises a valuable point. In our case, the Venturi effect is not exactly what we believe is acting here, because this refers to the speed-up and pressure drop that occurs through a narrow passage with solid boundaries. However, we do agree with the reviewer that there is a speed-up when the flow gets deviated by the iceberg, which also leads to pressure drops (Bernouilli's principle). The figure below displays the simulation results showing surface friction velocity and flow streamlines around the iceberg, with streamline color representing wind speed.

[Figure]

At this point we'd like to point out that one of the authors of this manuscript worked on extensive simulations on snow transport and deposition in the iceberg-carrying AtkaBay setting. The results of this work are, however, content of a standalone manuscript, which is under review. Therefore, we aim to address the reviewer's comment by adding a detailed suggestive statement that a speedup on the iceberg lee side—arising from both flow reattachment and turbulent momentum mixing, with their relative importance depending on atmospheric stability—is expected, leading to a very thin snow cover. However, we cannot go into too much detail about the simulation results in order not to reveal too much ahead of the other study.

3) The easterly wind direction is mentioned several times as the predominant wind direction. Can a wind rose be provided from the nearest weather station to support the Klöwer et al., 2013 study? Undoubtedly there are winds from other directions which can often produce secondary drifting patterns on the snowscape.

> This is a great suggestion, which we will gladly implement. We have access to wind speed and direction data from the meteorological observatory from Neumayer III Station, which is located ~ 6.5 km from the western fast ice edge of Atka Bay. A wind rose will be added to Figures 1c, 2, and 4 to better compare with the patterns on the ice berg lee sides. The wind rose values in these Figures will be from November 5$^{th}$ to December 5$^{th}$, hence covering one month. To further explore the wind patterns, we intend to add an additional appendix figure showing wind direction and speed patterns on a weekly basis between October 5$^{th}$ and December 5$^{th}$.

**Minor Comments:**

L256-258; L383-384; L420-422. While I generally agree with the statement that high backscatter snow covered sea ice corresponds to larger topographic roughness, one has to be careful in entirely associating this high backscatter with surface roughness (even though your surface roughness metric from the ALS DEM data suggest as much). For example, in the attached supplement I have uploaded, there is a small iceberg (highlighted in red box) which has high backscatter in the lee of the iceberg but does not show the high roughness in the center region of the lee (other than the lateral side edges as described by the authors). So, it apparently does not occur in all cases. In my opinion, it is equally likely that this thin snow region can permit the warmer air temperatures to produce higher basal snow layer temperature and brine volume, altering dielectrics and increase volume scattering (as you allude to in L279; L341-350 and elsewhere). In other words, it could be MORE than just surface roughness, especially for your Type 2 reflections. This process is nicely described in https://ieeexplore.ieee.org/abstract/document/9000883

[Figure]

RC1: 'Comment on egusphere-2025-2657', John Yackel, 14 Jul 2025

*Figure from the reviewer's supplement highlighting high SAR backscatter on the lee of an iceberg, which is unrelated to topographic roughness of the surface (red rectangles).*

**We will gladly implement the reviewer's suggestion and agree that not in all cases (however in the vast majority of cases) the topographic roughness of the surface is most likely responsible for high backscatter in SAR images. The suggested alternative mechanism (warmer air temperatures leading to warmer snow, higher brine volume and therefore altering the dielectric properties) as well as the suggested reference are a valuable addition to this topic in the manuscript. However, they do not seem to apply on most cases of the thick snow drifts that generally have lower backscatter.**

Table 1 - is it possible to provide AFIN drill site labels for Figure 1 circles?

**Agreed. The drill site labels will be added in Figure 1c.**

L335 ... typo 'single'

**Will be corrected.**

L358 .. while snow-ice formation horizon is a likely candidate, a brine-wetted snow snow layer and its effect on dielectric properties and scattering, owing to the warm air and snow temperatures, is equally likely.

**We agree and will modify the text accordingly.**

L369 .. typo ... space needed

**Will be corrected.**

---

## Author Comment (AC2)

RC2: 'Comment on egusphere-2025-2657', Anonymous Referee #2, 08 Sep 2025

The manuscript 'Iceberg influence on snow distribution and slush formation on Antarctic landfast sea ice from airborne multi-sensor observations' by Franke et al. illustrates how grounded ice bergs on landfast Antarctic sea ice affects snow redistribution on the windward side of the icebergs that induces thick snow drift accummulation and flooding. This is observed from a multi-platform and scale campaign on Atka Bay from December 2022 using UWBM Radar, ALS, NIR imagery, ground-based validation and imaging SAR. The paper is well-written, however, the theoretical 'treatment' and 'handling' of radar scattering mechanisms (mostly speculated and neglectance of double-bounce scattering), in combination with assumption of snow only as dry and flooded snow, makes the paper slightly speculative and needs considerable improvement. Therefore, I recommend major revisions for this round. My comments are general for this round and do not deal with specific line-by-line comments for now. The paper is structually well-written with outstanding depiction and quality of figures that may require some change once my feedback is incorporated or defended.

> **We thank the reviewer for the time and effort reviewing our manuscript as well as for the helpful feedback and comments with respect to material properties and scattering interpretation. We outline below how we intend to address the comments and how we will modify our manuscript accordingly.**

General Comments:

1) My major concern is interrelated. a) The paper assumes radar reflections originating '**either the transition from snow to ice or the transition from dry snow to flooded snow, which could be wet snow / slush or refrozen, salty snow ice**.' Now our issue is this. When the thick snow on thinner ice induces negative freeboard, yes, slush and refrozen snow-ice forms. But, what the authors missed (surprising) is that slush formation during water percolation into the snow also causes snow layers above the slush to be highly saline and completely brine-wetted. This is not 'dry-snow' especially at air and snow temperatures between -4 and -1C. The authors in Line 220 mention '**we specifically define our snow column thickness as dry-snow thickness. The dry-snow depth represents either the interface between snow and ice or the interface between dry snow and flooded snow (i.e., slush).**' This treatment of 'snow column' neglects this intermediate but highly scattering layer of this brine-wetted snow volume, highly sensitive to radar waves, even before the scattering originates from the slush layers below. This needs to be accounted for and addressed.

> **We thank the reviewer for the feedback and agree with the reviewer's concerns regarding point 1). We will expand the simplification of our materials (dry snow and slush) and include the effect of water percolation in snow, as well as its impact on the scattering properties in the radargrams. We will add**

**references covering work by Nandan, Mallet, and others in that regard and will include a more extensive discussion. We will change our two-layer system (snow and slush) to a three layer system (snow, brine-wetted snow and slush). We agree that the term dry snow may be misplaced.**

b) My second concern is tied to my first concern above. The paper's treatment of scattering mechanisms depicted in Figure 3, sections 4.2 and 5.2, figure 5 is very speculative and vague. First of all, thick snow on Antarctic sea ice, exhibiting dirunal air and snow temperature changes during the spring season leads to constant microstructural variations vertically, with formation of melt-refreeze layers, snow grain metamorphism and most importantly (in this study context), changes in snow brine volume from both the brine-wetted layers (omitted in the study) and slush layers underneath. This significantly modifies the snow/slush dielectrics leading to ambigious penetration of radar waves. Next, the paper uses reflection and scattering side by side which needs to be corrected. Figure 5 for example uses snow reflection from surface, volume and interfaces, but they are scattering processes than reflection. Next, with an iceberg sitting grounded on sea ice, double bounce scattering is the major contributor from SAR imagery which also needs to be addressed. Next, the theoretical interpretation of scattering mechanisms needs some modeling evidence to be conclusive than speculative. But I also understand there are no in situ samples of snow and slush from the site (or do you?). I understand the logistic difficulties to collect samples from close to an iceberg due to risk of its tipping. But I feel, irrespective of that, there are slush studies from past that could be used to 'less speculatively' interpret the role of snow and slush on radar signatures. I also strongly suggest to remove 'dry snow' from the paper as the temperatures are well above the dry snow thresholds.

**We also thank the reviewer for this comment, which is very helpful for improving the paper. Addressing point 1, we will revise the complexity of the snowpack. Additionally, we will clarify the use of the terms "scattering" and "reflection" throughout the text. We are aware of double bouncing as a suggested mechanism for lake ice scattering by bubbles near the bottom of lake ice, but believe that it is of limited significance here (which we will discuss and justify in the revised version) as scattering in snow is much more complicated.**

**Indeed, there are no in situ samples from the regions where we detected slush, and the reviewer is correct in assuming that these could not be collected due to the proximity to icebergs. We could only obtain a few samples from the level snow away from icebergs. Based on the results of this study, however, we believe that in situ samples from these regions would be highly valuable and should be included in future research.**

**Regarding the suggestion to incorporate modeling studies and additional literature references on slush over sea ice, we will review the relevant literature and cite it where appropriate. We will also remove the term "dry snow" from the paper and replace it with the term "fresh snow", which we will define to be most likely non-saline and not brine-wetted. However, we consider robust modeling the snow response to radar waves beyond the scope of our study but agree that such modeling represents promising future directions.**

c) Assumption of snow densities: The paper assumes an average snow density of 0.38 ± 0.06gcm-3. Well, with such a flooded snow volume, the assumption of 0.38 is highly underestimated. Is it possible to conduct a quick sensitivity analysis to use both dry snow density and slush density from literature to average them and redo the snow thickness calculation (provided you consider slush also as a part of the snow volume)? Also, Figure 4, the classified imagery is also not interpretive unless value ranges are shown. For example, SAR backscatter showing 'low' and 'high' values can be of any range correct? unless we show it. Also, instead of classifying scattering mechanisms can Types 1 to 3, I suggest to use scattering from geophysical interfaces/volume (e.g. air/snow, within-snow, slush etc etc), as using Types makes it difficult for a future author to use them as a proper terminology. Makes sense?

These are my major comments for now. Like I said, the paper is novel but needs considerable changes with respect to my concerns. I am sure the manuscript can be considerably improved if you think about these changes (or defend).

**We agree with the reviewer that the snow density for snowpacks containing slush is underestimated. However, we would like to point out that the effect of the higher density of slush is minor, as the top surface of the slush already represents the reflection, and we expect little backscatter within the slush due to strong attenuation. We would therefore disagree with the reviewer in the sense that—as the reviewer points out—we do not consider slush as part of the volume through which the radar waves we measure propagate, except possibly for a few centimeters, i.e., the volume where depolarization occurs. Instead, we would argue that the regions where we detect slush also exhibit greater snow thickness. The reference measurements in the literature were conducted for snow thicknesses of less than one meter, whereas the regions where we detect slush have snow thicknesses of several meters. This results in greater compaction and, consequently, increased density. As a result, snow thickness is overestimated in these areas—a point we already address in line 291, which also explains the apparent downbending of the reflection in Figure 5c.**

If we were to use the average density of dry snow and slush for the entire study area, as suggested by the reviewer, we would underestimate snow thickness in areas without slush, which constitute the majority of the study area. In the revision of our article, however, we will discuss the issue of higher density in regions with slush in greater detail.

For Figure 4, we will display the full value ranges of the colormaps, and we agree with the reviewer that we should have done so from the beginning.

We also agree with the reviewer that it is more appropriate to rename the classification of Types 1–3 as geophysical interfaces/volumes. Thank you for raising this point.